# Comparison of Aerial Thermal Infrared Imagery and Helicopter Surveys of Bison (*Bison bison*) in Grand Canyon National Park, USA

**DOI:** 10.3390/s21155087

**Published:** 2021-07-27

**Authors:** Jacob D. Hennig, Kathryn A. Schoenecker, Miranda L.N. Terwilliger, Gregory W. Holm, Jeffrey L. Laake

**Affiliations:** 1Contractor with the U.S. Geological Survey, Fort Collins Science Center, Fort Collins, CO 80526, USA; 2U.S. Geological Survey, Fort Collins Science Center, Fort Collins, CO 80526, USA; schoeneckerk@usgs.gov; 3National Park Service, Grand Canyon National Park, Grand Canyon, AZ 86023, USA; Miranda_Terwilliger@nps.gov (M.L.N.T.); Gregory_Holm@nps.gov (G.W.H.); 4Independent Researcher, Escondido, CA 92025, USA; jefflaake@gmail.com

**Keywords:** abundance estimation, aerial survey, double-observer, sightability, residual heterogeneity

## Abstract

Aerial thermal infrared (TIR) surveys are an attractive option for estimating abundances of large mammals inhabiting extensive and heterogeneous terrain. Compared to standard helicopter or fixed-wing aerial surveys, TIR flights can be conducted at higher altitudes translating into greater spatial coverage and increased observer safety; however, monetary costs are much greater. Further, there is no consensus on whether TIR surveys offer improved detection. Consequently, we performed a study to compare results of a TIR and helicopter survey of bison (*Bison bison*) on the Powell Plateau in Grand Canyon National Park, USA. We also compared results of both surveys to estimates obtained using a larger dataset of bison helicopter detections along the entire North Rim of the Grand Canyon. Observers in the TIR survey counted fewer individual bison than helicopter observers (101 to 127) and the TIR survey cost was 367% higher. Additionally, the TIR estimate was 18.8% lower than the estimate obtained using a larger dataset, while the comparative helicopter survey was 9.3% lower. Despite our small sample size, we found that helicopter surveys are currently the best method for estimating bison abundances in dense canopy cover sites due to ostensibly more accurate estimates and lower cost compared to TIR surveys. Additional research will be needed to evaluate the efficacy of these methods, as well as very high resolution satellite imagery, for bison populations in more open landscapes.

## 1. Introduction

Accurate estimates of wildlife abundances are critical for implementing effective management strategies. Abundance estimates of species inhabiting extensive areas with heterogeneous terrain are routinely derived from aerial survey counts, but these surveys are expensive and have inherent risk factors. Consequently, optimizing monetary resources, personnel safety, and effectiveness of aerial surveys has challenged management agencies for decades [1]. Additionally, survey performance and success vary with habitat type and percent tree cover [2]. A universal study design or survey type does not exist, so individual management agencies must evaluate tradeoffs when deciding which method to employ to best match budget constraints, habitat type, or management objectives. Integrating remote sensing technology into aerial survey methodology has emerged as a potential way to improve animal detection [3], but direct comparison to standard protocols is needed before management agencies can fairly evaluate this newer technology.

Thermal infrared imagery (TIR) sensors detect differences in heat signatures between animals and background radiation [4]. Concordantly, TIR sensors attached to aerial platforms may increase detection of vertebrates in landscapes with heterogeneous topography or vegetation cover compared to human observers operating solely within the visible color spectrum [5,6,7,8]. Further, these sensors have the ability to detect animals at greater distances than humans, which allows for higher flight altitudes compared to helicopter or fixed-wing surveys [9]. Conducting surveys at higher altitudes translates into the potential for increased spatial coverage and decreased risk for human pilots and observers onboard the aircraft [9,10].

While the benefits of aerial TIR surveys are apparent for safety, there remains no consensus on whether they perform better in animal detection than human observers in traditional helicopter or fixed-wing flights [11,12]. For instance, in wooded habitats, TIR sensors had higher detection rates of kangaroos than human observers, but this result reversed in open areas [12]. Additionally, the cost of contracting commercial entities with high-resolution cameras needed to perform surveys is usually greater than for human observer helicopter flights (see later). Consequently, an evaluation of survey effectiveness and cost between TIR and traditional methods can assist management agencies facing a decision on which method to employ for their specific needs.

Public land managers in charge of managing bison (*Bison bison*) on the Kaibab Plateau in northern Arizona are currently debating such a decision. The range of this population encompasses both the Kaibab Plateau National Forest, managed by the U.S. Forest Service (USFS), and Grand Canyon National Park (GCNP), managed by the National Park Service (NPS). On USFS lands, wildlife, including bison, is managed by the state of Arizona through the Arizona Game and Fish Department (AFGD), while the wildlife on NPS lands is not. These three agencies differ in their respective views regarding this bison herd, stemming from this herd’s history and the stated missions of each agency. This herd was introduced to the region, and uncertainty remains over whether the area was within historical range of bison [13]. The AFGD manages bison hunting on USFS land and is invested in sustaining a huntable population within the Kaibab National Forest [13]. Subsequently, hunting pressure results in bison refuging on NPS land, where they are protected from hunting, resulting in the degradation of other resources that the NPS protects [14]. Indeed, bison grazing has been linked to degraded riparian habitat within the park [14]. All agencies agree that too many bison are currently on the park landscape; they are actively working to reduce the population through various means [15,16]. Consequently, an accurate population estimate is critical to determining the success of herd-reduction activities.

A robust examination of herd size is lacking and difficult to obtain given the rough topography, tree canopy cover, and heterogeneous landcover types. An accurate estimate of the Kaibab bison population would be useful to guide the aforementioned agencies in determining how to appropriately implement management actions, such as lethal removal. Current survey procedures involve using a helicopter to conduct double-observer sightability (DOS) surveys [17]. Aerial TIR imagery has been proposed as an alternative that can both improve detection and increase observer safety compared to helicopter-based flights. We adapted the DOS design for use in an aerial TIR survey and compared model-adjusted counts between TIR and helicopter surveys of bison on the Powell Plateau within Grand Canyon National Park. We also compared how each method performed relative to an ostensibly more accurate estimate obtained by using a larger dataset of helicopter detections. Our objective was to directly compare results from TIR and helicopter-based DOS surveys of a large mammal conducted on the same day within a heterogeneous landscape.

## 2. Materials and Methods

We conducted aerial surveys of bison on the North Rim of Grand Canyon National Park, Arizona, USA (Figure 1). Due to budget limitations, we carried out comparison surveys of TIR and helicopter flights solely on the Powell Plateau, an area indicated by park biologists to contain the greatest concentration of bison in winter (Figure 1). Powell Plateau is approximately 30 km^2^ and ranges in elevation from 1910–2240 m [18] (USGS 2020). Landcover on the plateau contains both dense-canopy ponderosa pine (*Pinus ponderosa*) forest and relatively open-canopied pinyon pine (*Pinus edulis*) and juniper (*Juniperus* spp.) woodland.

We conducted the comparison TIR and helicopter surveys on 11 February 2020 [19]. We timed the surveys to coincide with heavy snowfall, which increases contrast between bison and immediate surroundings in both in the color and infrared spectrum and thereby theoretically increases detection. We staggered the flight times of the surveys to avoid complications with two aircraft simultaneously in the air, but also to optimize conditions for each type of survey: TIR in the early morning hours when thermal contrast is highest, and DOS at high noon to reduce shadows for visual observers. The TIR flight occurred between 0700–0830 local time, and the helicopter flight occurred between 1130–1240. We also conducted helicopter surveys of the entire herd along the North Rim of the Grand Canyon on 25 March 2019, 11 February 2020, and 1 February 2021 [19].

The helicopter survey followed DOS protocols described in Schoenecker et al. (2018) and Griffin et al. (2020) [20,21]. We conducted the survey using a helicopter (Bell 206BIII) with a 4-person crew including a pilot and three observers (one frontseat, two backseat). The helicopter flew predefined transects spaced 800 m apart at 60 m above ground level (AGL). To ensure independence of detection, observers were not allowed to communicate with one another during the flight until the aircraft passed abeam of a bison group [20]. Once this occurred, crew members discussed which observer(s) detected the bison group and worked together to record sighting covariates including vegetation cover type, whether the group was in motion when detected, percent concealing vegetation and percent snow cover within 10 m of the group [21]. The pilot circled detected groups to help observers obtain an accurate count and to facilitate photographing groups for post-flight confirmation of group size.

We contracted Owyhee Air Research, Inc. (Nampa, ID, USA) to conduct the aerial TIR flight using a fixed-wing aircraft with a 360° pan-tilt-zoom (PTZ) gimbal thermal infrared camera (640 × 512 IR resolution, L3Harris MX-10, Melbourne, FL, USA) mounted to the bottom. The fixed- wing aircraft flew predefined transects at approximately 750 m AGL. The camera was kept at a 45–50° down-angle from horizontal and was subject to change due to small variances in landscape topography and aircraft altitude. The camera relayed real-time video to two separate monitors inside the aircraft cabin where two observers acted independently to detect and count bison in a DOS design. When a detection occurred, observers increased camera zoom and switched between TIR and natural color views to count the number of individuals per group. Cameras recorded video of each detection for post-flight accuracy assessments.

Prior to the surveys, we radio-marked bison in the population with telemetry collars, which allowed us to account for residual heterogeneity in detection probability [2,22,23,24]. We deployed 25 global positioning system (GPS) collars (Telonics Inc., Mesa, AZ, USA) on bison in fall 2019 (*n* = 14) and fall 2020 (*n* = 11). We captured bison using corral bait trapping; animals were held in a hydraulic squeeze chute for collar application on both occasions. Observers in the helicopter survey recorded if detected groups contained an individual wearing a neck collar, but the TIR survey was conducted at too high an elevation for observers to see collars even with a camera zoom feature. Post-flight assessment determined that the videos and photos taken during each survey were not of high enough resolution to conclusively determine which of the detected groups contained a collared individual. Consequently, we compared timestamps and locations between survey detections and GPS-collar location fixes to assess which collared groups were detected by each survey method.

### Statistical Analysis

We fit detection models with the Huggins [25,26] closed-capture estimator using the ‘RMark’ [27] (Laake 2013) and ‘DoubleObs’ [28] packages within Program R Version 4.0 [29]. For all analyses, we included an additional observer in our model structure corresponding to detections of radio-marked groups. For the helicopter surveys, we pooled detections by the pilot and frontseat observer as the first observer, and both backseat observers as the second observer. Thus, all models contained three capture histories: radio observer, observer 1, and observer 2. We fixed detection probability of the radio observer to *p* = 1.0 for groups containing a telemetry collar and *p* = 0.0 for groups without a telemetry collar. For each dataset, we first assessed which of three residual heterogeneity models best fit the data. These models estimated residual heterogeneity as: (1) the difference in detection probability between radio-collared and uncollared groups [22], (2) a function of recapture probability [25], or (3) a model combining both approaches. We then compared the top-ranked heterogeneity model to an observer-only model that does not account for residual heterogeneity. We did not include sighting covariates for the comparison TIR and helicopter surveys due to limited sample size, but we did include the effect of forested cover type and the natural logarithm of group size in models of the pooled detection dataset. We used Akaike’s Information Criterion corrected for small samples sizes (AIC_c_) [30] to rank models and calculate model-averaged abundance estimates using all models. Regarding the pooled dataset, we only calculated abundance estimates from detections occurring on Powell Plateau on 11 February 2020.

## 3. Results

Aerial TIR observers detected 15 bison groups with only one missed group between the observers: a single bison detected by the first observer (Table 1). The helicopter crew detected 13 groups with backseat observers detecting all 13 groups and the frontseat observers detecting four groups. Aerial TIR observers counted a total of 101 bison across groups compared to 127 counted by helicopter observers (Table 1). The pooled dataset from the three helicopter surveys along the entire North Rim included 95 detections.

During the comparative TIR and helicopter surveys, five bison groups on Powell Plateau contained a telemetry-collared individual. Both the helicopter and TIR surveys detected four of the five collared bison groups, but each missed a different group. The collared group that the TIR survey missed was found by the helicopter observers to contain 28 bison, while the group that the helicopter observers missed was found by TIR observers to contain eight bison.

The model accounting for residual heterogeneity as the difference in detection between collared and uncollared groups best fit the TIR dataset, while estimating residual heterogeneity through recapture probability best fit the helicopter datasets; however, models not accounting for residual heterogeneity best fit the Powell-only datasets (Table 2).

Regarding the pooled dataset, the natural logarithm of bison group size was the top-supported sighting covariate (*w_i_* = 0.71) with detection increasing with larger group sizes (Table 2; Figure 2). Presence of forested cover type decreased detection probability but had little model support (*w_i_* = 0.30; Table 2; Figure 2). Model-averaged abundance estimates for bison on Powell Plateau on 11 February 2020 were lower for the TIR survey compared to the helicopter survey, and both were lower than estimates generated from the pooled helicopter dataset (Table 3).

## 4. Discussion

Our study revealed greater raw counts and estimated abundances of bison on Powell Plateau by the helicopter survey using human observers compared to the TIR survey. True abundance was not known so we evaluated the accuracy of both survey estimates relative to estimates generated from pooled detections across three helicopter surveys. Estimates from the TIR survey were 18.8% lower than the pooled estimate, whereas helicopter estimates were 9.3% lower. The estimate generated by the pooled dataset was not within the 95% confidence intervals of the TIR estimate, but it was for the helicopter estimate.

The lower raw count and model-generated estimates of the TIR survey is primarily attributed to a missed group of 28 bison that was detected by helicopter observers but not the TIR survey. This group was located in an area where solar radiation was high, and TIR observers reported strong “thermal clutter” that made the background habitat appear white instead. Animal detections in TIR surveys are made by spotting the white heat signatures against dark backgrounds; thus, strong thermal clutter confounds the heat signatures of animals and reduces the ability of observers to detect them. This was not an issue for helicopter observers using the visible color spectrum. A nocturnal TIR flight would likely decrease the influence of thermal clutter and theoretically increase detection [12]; however, a night flight introduces extra safety concerns including reduced situational awareness and increased risk of collision [9,12]. Degree of thermal clutter could possibly be used as a sighting covariate for TIR surveys in a DOS design; however, for our TIR survey, this covariate could not be determined for missed marked groups.

Both TIR observers were experienced and had nearly identical bison detections. This could be because of observer training, but more likely it is a function of each observer having identical viewsheds in the TIR survey (two monitor screens, but only one camera view). Since both observers saw exactly the same view of the survey area on their monitors, their results closely matched. Conversely, we observed a large difference in detection between helicopter observer positions. We attribute this difference to the larger field of view for the first observer (pilot and frontseat observer), in which the frontseat observer has to scan a wider area to the front and side of the helicopter, while observer 2 (backseat observers) search a narrower viewshed on the side of the helicopter and a farther distance out. Further, the first observer included detections by the pilot, who was mainly concerned with flying the aircraft, while both backseat observers could fully focus their attention on finding bison. Bison group size and presence of forest cover influenced detection in helicopter surveys, and these sighting covariates could potentially be added to future TIR surveys combined with larger sample sizes to improve detection estimates.

We designed our study to account for residual heterogeneity in detection probability among bison groups, which is critical for reducing bias in double-observer wildlife surveys even in designs that account for imperfect detection [31,32]. Using detections of bison on Powell Plateau only, models incorporating residual heterogeneity received comparatively less support than observer-only models. This was likely due to the inclusion of an extra parameter along with little information about residual heterogeneity with only one missed group. Conversely, heterogeneity models were better fits for the larger, pooled dataset, signifying that accounting for heterogeneity was critical. Our study design relied on using inclusion of marked and unmarked groups to account for imperfect detection and residual heterogeneity, but neither the TIR nor helicopter observers could conclusively detect whether a bison group contained a collared individual (collars matched bison pelage color). We addressed this by comparing timestamps of detections and GPS locations of bison to match seen and unseen collared groups. This is not an ideal approach for future surveys because GPS fix rates were infrequent (2 h) meaning bison location fixes were not completely synchronized with timing of the helicopter flyover, and without having visual observation of the missed radio marked group, we were unable to assess sighting covariates for those groups. Conducting a fixed-wing telemetry reconnaissance flight immediately preceding helicopter and TIR surveys has been used successfully to determine location and sighting covariates of all collared groups [2]. We did conduct such a telemetry flight in 2020, but observers were not able to locate all collared groups and therefore did not obtain GPS locations, group sizes, and sighting covariates. We rectified these errors before conducting the telemetry flight preceding the 2021 helicopter survey.

The helicopter survey with onboard observers found more bison (raw count) as did the corrected estimate (detection models) than the TIR survey. Our sample size was very limited, in part because the cost of TIR surveys is threefold higher than helicopter surveys. The helicopter survey was approximately $6,000 USD, whereas the TIR survey was $22,000 USD for the same test area on Powell plateau. This extrapolates to $14,000 for a helicopter flight and $50,000 for a TIR survey for the entire bison population across the North Rim. A key benefit to TIR surveys is the lower relative safety risk compared to manned helicopter or fixed-wing surveys. Another option that entirely removes the risk of human injury or death and would also decrease survey cost is to use very high resolution (VHR) satellite imagery to count bison. Using VHR to remotely assess vertebrate populations is an increasingly common method [33,34,35], but three main criteria must be met for VHR to be useful in generating abundance methods: large animal size, organism contrast with background landscape, and open terrain [36]. Bison populations in the North American Great Plains (i.e., Badlands and Wind Cave National Parks), satisfy these criteria; however, the dense canopy cover in our study area makes this technology unsuitable for this herd. Thus, park managers will need to decide if the price of TIR imagery is worth the safety benefits given the lower detection observed in our study. Testing whether alterations to TIR surveys could reduce the influence of thermal clutter and yield more promising results would be useful. Given the higher raw count, ostensibly more accurate estimates, and much lower cost, we conclude that helicopter flights with human observers are currently the best method for estimation and monitoring of bison abundance in Grand Canyon National Park.

## Figures and Tables

**Figure 1 sensors-21-05087-f001:**
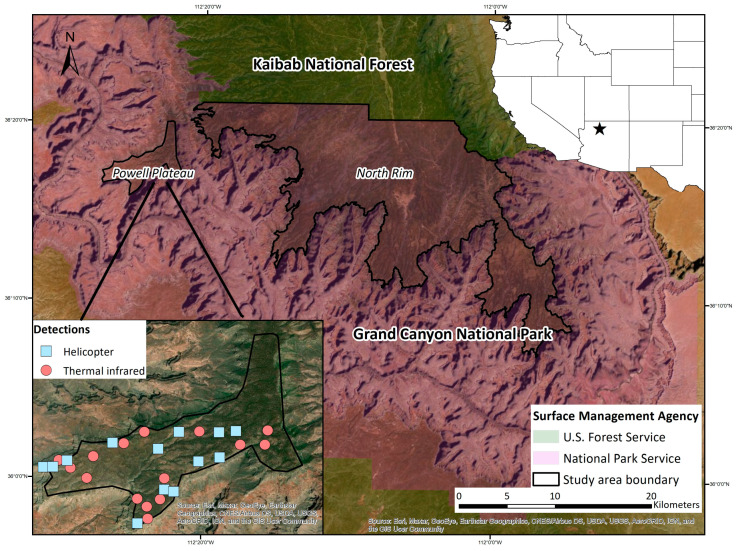
Location of study area and surface management agencies, Powell Plateau, and North Rim of Grand Canyon National Park, USA. Inset displays the location of bison group detections made by helicopter and thermal infrared surveys on Powell Plateau, 11 February 2020.

**Figure 2 sensors-21-05087-f002:**
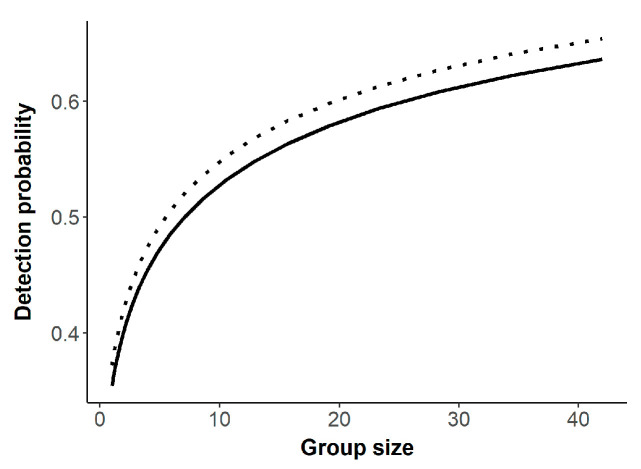
Detection probability as a function bison group size and presence of forested landcover type (solid = forested, dotted = not forested), North Rim of Grand Canyon National Park, USA, 2019–2021.

**Table 1 sensors-21-05087-t001:** Tally of raw counts of bison groups detected and individuals counted by each observer between independent helicopter and thermal infrared surveys, Powell Plateau, Grand Canyon National Park, USA, 11 February 2020.

Observer	Groups Detected	Individuals Counted
Helicopter-frontseat observers	4	38
Helicopter-backseat observers	13	127
Thermal infrared-observer 1	15	101
Thermal infrared-observer 2	14	100

**Table 2 sensors-21-05087-t002:** Model-selection results of bison detection probability from three different datasets: a thermal infrared survey on Powell Plateau, a helicopter survey on Powell Plateau, and a pooled dataset from three helicopter surveys of the North Rim of Grand Canyon National Park, USA. Columns represent the number of parameters (*K*), Akaike’s Information Criterion adjusted for small sample sizes (AIC_c_), difference between a given model and the top-ranked model (ΔAIC_c_), and the model weight (*w_i_*).

Model	*K*	AIC_c_	ΔAIC_c_	*w_i_*
*Thermal infrared-Powell*				
Observer ^a^	2	23.62	0.00	0.58
Observer + type ^b^	3	24.25	0.63	0.42
*Helicopter-Powell*				
Observer	2	26.98	0.00	0.59
Observer + recapture ^c^	3	27.68	0.70	0.41
*Helicopter-Pooled surveys*				
Observer + recapture + groupsize ^d^	4	200.10	0.00	0.45
Observer + recapture + groupsize + forest ^e^	5	201.74	1.64	0.20
Observer + recapture	3	201.88	1.78	0.18
Observer + recapture + forest	4	203.63	3.53	0.08
Observer + groupsize	3	204.98	4.88	0.04
Observer	2	205.86	5.75	0.03
Observer + groupsize + forest	4	206.70	6.60	0.02
Observer + forest	3	207.72	7.62	0.00

^a^ Observer 1 vs. 2; ^b^ Residual heterogeneity estimated as detection difference between marked and unmarked groups; ^c^ Residual heterogeneity estimated through recapture probability; ^d^ Natural log of group size; ^e^ Forest cover type.

**Table 3 sensors-21-05087-t003:** Raw counts, model-averaged abundance estimates, standard errors, and 95% confidence intervals for bison abundance on Powell Plateau, 11 February 2020. Models were generated from three datasets: a thermal infrared survey on Powell Plateau, a helicopter survey on Powell Plateau, and a pooled dataset from three helicopter surveys of the North Rim of Grand Canyon National Park, USA.

Method	Count	Estimate	SE	LCL	UCL
Thermal infrared-Powell	101	131	4	130	155
Helicopter-Powell	127	147	20	137	283
Helicopter-Pooled surveys	127	162	24	141	258

## Data Availability

We have made the dataset and associated metadata files available through a U.S. Geological Survey data release: https://doi.org/10.5066/P9CXDC8V (accessed on 24 June 2021).

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
