# Peer review of "Comparison of Aerial Thermal Infrared Imagery and Helicopter Surveys of Bison (Bison bison) in Grand Canyon National Park, USA"

_sensors, 2021, doi:10.3390/s21155087_

Round 1

Reviewer 1 Report

Henning et al present a comparison study of bison surveys via aerial TIR surveys and visible light surveys.  I find the presentation exceptional and the results and conclusions nicely follow from the recommendations.  I would like to know more about thermal cluttering (for example, please add another example).  I would also like to see the authors comment on the feasibility of adding a second thermal camera with a different point of view.  Was the thermal camera pointed at nadir?

Here are a few very minor comments:

Line 69 is restoral the word you want to use here?

Line 258 suggest deleting "a"

Line 294 change thanks to thank

Author Response

Reviewer 1

Hennig et al present a comparison study of bison surveys via aerial TIR surveys and visible light surveys.  I find the presentation exceptional and the results and conclusions nicely follow from the recommendations.  I would like to know more about thermal cluttering (for example, please add another example).  I would also like to see the authors comment on the feasibility of adding a second thermal camera with a different point of view.  Was the thermal camera pointed at nadir?

Response: Thanks, we very much appreciate the compliments. In regards to your comments, we have included more text to better detail what thermal cluttering is and how it affects detection. We also mention in that the thermal camera was a 360 PTZ gimbal and was oriented 45-50° below horizontal. We refrain from discussing the addition of another camera as this would increase cost and would introduce more complexity in analysis. We also believe that one camera is sufficient for covering the study area given our transect spacing.

Here are a few very minor comments:

Line 69 is restoral the word you want to use here?

Response: We meant to use the word “restore”. Thank you for catching this error.

Line 258 suggest deleting "a"

Response: We changed the text in this sentence to better reflect our intended meaning, it now reads: “Conducting a fixed-wing telemetry reconnaissance flight immediately preceding helicopter and TIR surveys…”

Line 294 change thanks to thank

Response: We have made this change.

Reviewer 2 Report

After careful review of the manuscript, I found its methods sound and its results valuable for future monitoring of the species. I would only recomend to detail more the differences between the methos in terms of survey conditions, altitude, and areas covered. Including part of this information as effort tracks and groups detected in an additional map of results would make this more clear for the reader.

Author Response

Response: Thank you, we have included an inset within Figure 1 displaying the locations of detections made by both the thermal infrared and helicopter surveys to better depict that both surveys covered the same area on Powell Plateau.

Reviewer 3 Report

General comment

Authors produced a very interesting and useful paper comparing two aerial methods to monitor bison populations. Their analyses are robust. It is recommended for publication after some modifications.

Specific comments

Line 24 – Add a sentence in the Abstract about the limitations of the study, as recognized in the Discussion, and the need for further research and validation.

Lines 39-40 – Change to either “comparisons… are” or “comparison… is”.

Line 123 – Correct to “a thermal”.

Line 175 – Correct to “comparison of TIR”.

Lines 160-161 – Explain which models you used for model averaging. All models, models summing up to 95% of weight, models with ΔAIC < 2?

Line 196 – It is not obvious what this weight represents. The reader will go to Table 2 and see w = 0.45 for group size. Then someone have to remember that you did model averaging and then will have to sum all weights of models including group size. Please clarify this. Adding a Table including all model predictors with averaged weights would be of great help to the reader.

Table 3 – You surely state on lines 161-163 that you used only Powell Plateau detections from the North Rim surveys, but this is not obvious in Table 3. Someone will think that you compared Powell Plateau with North Rim detections, something that is wrong and you did not do it. Please clarify this in the Table 3 caption.

Lines 258-260 – This sentence is not clear. Please clarify.

Author Response

General comment

Authors produced a very interesting and useful paper comparing two aerial methods to monitor bison populations. Their analyses are robust. It is recommended for publication after some modifications.

Response: Thank you very much.

Specific comments

Line 24 – Add a sentence in the Abstract about the limitations of the study, as recognized in the Discussion, and the need for further research and validation.

Response: Good point, we now mention our limited sample size in the Abstract, and include a last sentence of, “Additional research will be needed to evaluate the efficacy of these methods, as well as very high resolution satellite imagery, for bison populations in more open landscapes.”

Lines 39-40 – Change to either “comparisons… are” or “comparison… is”.

Response: We changed to “comparison…is”.

Line 123 – Correct to “a thermal”.

Response: We have made this change.

Line 175 – Correct to “comparison of TIR”.

Response: We have changed to “comparative TIR…” to better reflect our meaning.

Lines 160-161 – Explain which models you used for model averaging. All models, models summing up to 95% of weight, models with ΔAIC < 2?

Response: We have updated the text to explicitly state that we used all models in model-averaging.

Line 196 – It is not obvious what this weight represents. The reader will go to Table 2 and see w = 0.45 for group size. Then someone have to remember that you did model averaging and then will have to sum all weights of models including group size. Please clarify this. Adding a Table including all model predictors with averaged weights would be of great help to the reader.

Response: We have added extra text to indicate that the weight represents the total model weight of all models containing this covariate; however, because we only evaluated 2 sighting covariates, we do not think that another table is required.

Table 3 – You surely state on lines 161-163 that you used only Powell Plateau detections from the North Rim surveys, but this is not obvious in Table 3. Someone will think that you compared Powell Plateau with North Rim detections, something that is wrong and you did not do it. Please clarify this in the Table 3 caption.

Response: We used all detections on the North Rim to build a model and then used this model’s results to generate abundance estimates for the subset of detections that occurred on Powell Plateau on 11 February 2020. We recognize that this was not immediately clear in the text/tables, and have updated them to improve understanding.

Lines 258-260 – This sentence is not clear. Please clarify.

Response: Yes, we apologize for the lack of clarity in this sentence. We have now changed the sentence to “Conducting a fixed-wing telemetry reconnaissance flight immediately preceding helicopter and TIR surveys…”